# Oncosis is the predominant type of cell death in rhabdomyolysis following exertional heat stroke

**Chengcheng Li[1,2], Yang Liu[1,3], Handing Mao[1], Wenjun Yang[1], Shuyuan Liu[1], Yi Shan[1]***

**1** Department of Emergency Medicine, The Sixth Medical Center of PLA General Hospital of Beijing,
**2** Department of Critical Care Medicine, the Third Affiliated Hospital of Sun Yat-sen University, **3** School of Medicine, South China University of Technology

* nghicu@163.com

## Abstract

### Background

Rhabdomyolysis (RM), particularly heat exhaustion-associated rhabdomyolysis (ehsRM), is a significant clinical issue associated with high mortality and healthcare costs. However, the cellular death mechanisms remain incompletely understood. Oncosis, a form of passive cell death distinct from apoptosis, is characterized by cell swelling and triggered by ATP depletion. Additionally, porimin, a specific biomarker, can uniquely identify oncosis. This study aims to investigate the role and mechanisms of oncosis in both in vitro and in vivo models of ehsRM.

### Objective

This study aims to investigate the role and mechanisms of oncosis in both in vitro and in vivo models of ehsRM.

### Methods

In the in vitro study, 6-8-week-old male rats were subjected to treadmill exercise at an ambient temperature of (39.5±0.5)°C and relative humidity of 50%-60%, at a speed of 15 meters per minute until their core body temperature (Tc) reached 43.0°C to establish a heatstroke animal model. Skeletal muscle and blood samples from the gastrocnemius were collected for cytokine, biochemical, and histopathological analyses. Pathological findings revealed decreased muscle fiber density, structural disarray, swelling, degeneration, and hemorrhage. Ultrastructural analysis showed cell swelling, structural disarray, cytoplasmic vacuolation, mitochondrial swelling and degeneration, loss of cristae, and nuclear degeneration, indicating myocyte swelling and necrosis. Porimin, CytC, Bax, and caspase-1 expression increased, while Bcl-2 expression decreased. JC-1 staining indicated a decline in mitochondrial membrane potential and dysfunction. ATP levels decreased, and reactive oxygen species (ROS) production increased. In the in vivo study, HSKMC cells were subjected to 4 hours of heat shock at 43°C to establish a

**Data availability statement:** Location of the data: figshare DOI/Accession Number: https://doi.org/10.6084/m9.figshare.28357793.

**Funding:** The funding support from the Military Medical Innovation Project (18CXZ019) for this work is gratefully acknowledged. We confirm that the funders had no role in the study design, data collection and analysis, decision to publish, or preparation of the manuscript.

**Competing interests:** The authors declare that they have no known competing financial interests or personal relationships that could have appeared to influence the work reported in this paper.

heatstroke-induced rhabdomyolysis cell model. Electron microscopy revealed cell swelling, cytoplasmic vacuolation, mitochondrial swelling and degeneration, and nuclear swelling; late-stage (necrotic-like death) was characterized by nucleolar dissolution, nuclear fragmentation, chromatin condensation, and collapse of cytoplasmic structures. After 24 hours post-modeling, the proportion of double-positive cells (porimin+/PI+) and ROS levels significantly increased, as did porimin expression, while mitochondrial membrane potential and ATP levels significantly decreased. The proportion of Annexin V+/PI+ double-positive cells and caspase-3 levels showed no significant changes.

## Results

In both in vitro and in vivo studies, oncosis played a crucial role in ehsRM. Pathological and ultrastructural analyses demonstrated cell swelling, structural disarray, mitochondrial damage, and nuclear degeneration. Porimin, CytC, Bax, and caspase-1 expression increased, while Bcl-2 expression decreased. ATP levels decreased, and ROS production increased. In the in vivo study, the proportion of porimin+/PI+ double-positive cells and ROS levels significantly increased, while mitochondrial membrane potential and ATP levels significantly decreased. The proportion of Annexin V+/PI+ double-positive cells and caspase-3 levels showed no significant changes.

## Conclusion

Oncosis is predominant in ehsRM, involving mitochondrial dysfunction, ATP depletion, and oxidative stress.

## 1. Introduction

The previous studies have demonstrated that cell death [1,2], apoptosis [3], autophagy [4,5], and necroptosis [6] are the predominant pathological changes observed in EHS, frequently coexisting in experimental models of acute muscle injury [7]. However, these diverse forms of cell death fail to elucidate the mechanism underlying RM development subsequent to EHS.

Currently accepted mechanisms suggest that deficiency of ATP due to high intensity exercise and continuous muscle contraction could induce the dysfunction of $Na^+$-$K^+$-ATPase, causing subsequent activation of reverse mode $Na^+$/$Ca^{2+}$ exchanger resulting in an increased level of $Na^+$ and $Ca^{2+}$ in the cells, which leading to the retention of water and sodium in cells, swelling and necrosis of cells. The mechanism and morphological changes of RM exhibit striking similarities to those observed in oncosis.

Oncosis, a form of porimin-dependent non-programmed cell death, is caspase-independent, which is different from apoptosis [8–10]. The earliest discovery of oncosis was observed by the scientist von Reckling, who found that bone cells undergo swelling death [8]. In 1995, Majno observed cell swelling, blebbing, and karyolysis accompanied by an inflammatory reaction, hence the name tumor due to the swelling of cell morphology [11]. oncosis is linked to energy expenditure, resulting in the dysfunction of membrane ion pumps, dilatation of the ER and Golgi, mitochondrial condensation followed by their swelling, and formation of cytoplasmic blebs or blisters that are organelle-free [8,12].

In this study, we hypothesize that oncosis, plays a pivotal role in ehsRM. We constructed an ehsRM rat models and observed that the mechanisms underlying this condition include apoptosis, oxidative stress, and oncosis. Specifically, we hypothesized that oncosis plays a pivotal role in the development of rhabdomyolysis. To validate this hypothesis, we further established

a heatstroke-induced rhabdomyolysis model using HSKMC cells. Using Western Blot and RT-qPCR, we detected the upregulation of porimin, a specific marker for oncosis, at both the protein and mRNA levels. Additionally, flow cytometry analysis revealed an increased rate of porimin (+)/PI (+) cells, indicating enhanced oncosis.

Our results demonstrate that in EHS, the primary mode of skeletal muscle cell death is oncosis rather than apoptosis. Furthermore, we found that oncosis in skeletal muscle cells is closely associated with reactive oxygen species (ROS) production, mitochondrial membrane potential damage, and ATP depletion. These findings provide new insights into the cellular mechanisms of EHS-induced rhabdomyolysis and suggest potential therapeutic targets for managing this life-threatening condition.

## 2. Materials and methods

### 2.1. Animals

12 male SPF-grade SD rats weighing 180 ~ 200g were purchased from SPF Biotechnology Co., LTD. (Production license No.: SCXK (Beijing) 2019-0010).All the rats were accommodated for 1 week in the Specific Pathogen Free animal experiment center of Medical Discovery Leader. Rats were housed in a room maintained at 20 -24°C and humidity of $(50 \pm 5)\%$, under a 12 h day/night cycle. Both standard chow and sterile water were supplied at libitum. Then all rats underwent adaptive running training for 1 week, as shown in Table 1. All the experimental protocols have been approved by the Institutional Animal Ethics Committee of Kangtai medical inspection service in accordance with the Guide for the Care and Use of Laboratory Animals of the National Institutes of Health. (Approval Number: MDL2023-03-25-02)

### 2.2 Rat model of EHS with rhabdomyolysis

12 rats were were divided into control group (CN) and EHS group (EHS), with 6 rats in each group. The EHS group rats were placed in the climate chamber set to $39.5 \pm 0.5°C$ and 50- 60% relative humidity (RH) in the absence of food and water [13–15]. The rats were allowed to run on the treadmill at a constant speed of 15 m/min with 1 minute break every 5 minutes. The time point of at which the rats appeared to be exhausted and the Tc reach 43°C was taken as a reference point of EHS onset.Exhaustion is specifically manifested as follows: under painless prodding, the rats are unable to stand, have a movement score of 2 (able to perform full range joint movements such as righting reflex (+), limbs flexed, but unable to stand), wet fur, convulsions, lethargy, and even light coma. Immediately after the onset of EHS, the rats were then removed from the climate chamber, weighed, and returned to their original cages with an ambient temperature of $(22 \pm 2)°C$, RH of $(50 \pm 5)\%$ and with free access to food and water.

**Table 1. Intensity of 7-day running training in rats.**

|  | D1 | D2 | D3 | D4 | D5 | D6 | D7 |
|---|---|---|---|---|---|---|---|
| Speed (m/min) | 10 | 15 | 20 | 22 | 22 | 22 | 22 |
| Training time (min) | 15 | 15 | 15 | 15 | 20 | 25 | 30 |

Ambient Temperature 22 ± 2°C、Humidity 50% ± 5%

### 2.3 Cell lines and culture

Human skeletal muscle cells (HSKMC) were purchased from Wuhan Punosay Life Technology Co., LTD. The HSKMC cells were allocated into two groups: the control group (CN) and the heat stress group (HS). The control group was cultured in a cell incubator with 37°C and 5% carbon dioxide, while the cells in the heat stress group were subjected to heat stress in a hot water bath at 43 °C for four hours followed by incubation at 37°C and 5% carbon dioxide for another twenty-four hours.

### 2.4 Serum collection

The rats were anesthetized by intraperitoneal injection of sodium pentobarbital (50 mg/kg body weight) at 6 h after the onset of EHS. The time point of 6 h was chosen based on pretest. Blood samples were harvested, and serum was separated by centrifugation (Sigma, Germany 3-30K) at 3, 000 rpm for 10 min at 4°C, and then stored at -80°C for biochemical and enzyme-linked immunosorbent assay (ELISA) analysis.

### 2.5 Serum Mb and CK measurement

The levels of myoglobin (MB) was determined by the automated analyzer (China, XR220 Plus). Commercial ELISA kits (Jianglaibio, China, Shanghai)were used for measuring levels of Creatine kinase (CK) using a Microplate Spectrophotometer (THERMO, Nanodrop lite) according to the manufacturer's instruction.

### 2.6 Histological examination

All animals died spontaneously or were eventually killed by cervical dislocation 6 hours after EHS onset after anaesthesia with phenobarbital sodium and dissection for blood from the abdominal aorta, Samples of gastrocnemius was harvested immediately following blood withdrawal and preserved in 10% neutral buffered formalin. All the preserved tissues were paraffin embedded, sectioned, stained with hematoxylin and eosin (H&E) and examined microscopically (Leica DM3000, Wetzlar, Germany) in a blind way.

### 2.7 Western blotting

Gastrocnemius tissue or HSKMC cell homogenates were prepared to assess levels of caspase-1-p10, caspase-3, NLRP3, cytochrome C (Cyt-C), Bax, Bcl-2, and porimin. Proteins were extracted and concentration determination was carried out according to the BCA kit's guidelines. Proteins were separated by electrophoresis on a 10% polyacrylamide gel operating at a voltage of 80 V (8 V/cm). The proteins were then transferred to PVDF membranes, which were incubated with 5% BSA for blocking for 1 hour. Subsequently, membranes were incubated overnight at 4°C with the following primary antibodies: porimin (bs-7674R, Bioss, China), cytochrome-C (Cyt-C) (ab133504, Abcam, England), caspase-3 (bs-0081, Bioss, China), caspase-1-p10 (22915-1-AP, Wuhan Sanying, China), NLRP3 (bs-6655R, Bioss, China), Bax (50599-2-Ig, Wuhan Sanying, China), Bcl-2 (26593-1-AP, Wuhan Sanying, China), and β-actin (MD6553, MDL, China). After washing with TBST (3 times for 10 minutes each), membranes were incubated with secondary antibodies (MD91256, MDL, China) at room temperature for 1 hour, followed by TBST washes (3 times for 5 minutes each) for chemiluminescent detection. Illumination was performed using a gel imaging system (UVP, GelDoc-It310, USA). Target band grey value analysis was conducted using Image J software (ChemiScope6100, CLINX, China).

## 2.8. Immunohistochemistry

After dewaxing, the muscle tissue sections, prepared for immunohistochemical determination of porimin, caspase-3, caspase-1, and α2A-AR levels, were subjected to antigen retrieval in citric acid solution at high temperature and pressure for 20 minutes. Subsequently, endogenous peroxidase activity was blocked for 10 minutes, and the sections were then incubated with normal goat serum for 60 minutes at room temperature. After discarding the excess liquid, the sections were incubated with the following primary antibodies at appropriate concentrations in a humidified chamber at 4 °C overnight: porimin (1:100 dilution), caspase-3 (1:100), α2A-AR (1:100), and caspase-1 (1:200).The next day, the sections were washed with PBS (5 times for 3 minutes each), re-equilibrated at 37 °C for 40 minutes, and then incubated with a secondary antibody for 60 minutes. This was followed by three additional washes and DAPI counterstaining for 10 minutes in the dark to label cell nuclei. Subsequently, the sections were mounted with glycerin and immediately observed under a fluorescence microscope. Finally, the integrated optical density (IOD) of positively stained areas in the sections was analyzed using Image-Pro Plus software (ChemiScope6100, CLINX, China).

## 2.9 TEM

Muscle specimens and HSKMC specimens fixed in 1% osmic acid for 2 h, dehydrated with ethanol and Propylene oxide, soaked in Propylene oxide and 100% embedding solution and finally polymerized overnight at 37–60 °C to prepare 70 nm ultrathin sections. The sections were double-stained using 3% uranyl acetate and lead citrate, after which the ultrastructure of muscle was observed using JEM-1400Flash TEM (Olympus Optical Co., Ltd., Tokyo, Japan).

## 2.10 JC-1 fluorescence staining

Changes in Mitochondrial membrane potential (MMP)were determined using a JC-1 mitochondrial membrane potential assay kit (Beyotime, China) according to the manufacturer's instructions. All samples were estimated quantitation of fluorescence intensity using Image-Pro Plus software (ChemiScope6100, CLINX, China). The loss of MMP was reflected by the ratio of aggregates (red fluorescence) to monomer (green fluorescence).

## 2.11 RNA extraction and real-time PCR

Total RNA was extracted from the freshly harvested muscle tissues by using Trizol (invitrogen, MDL, MDL91201). According to the manufacturer's instructions (ABI-invitrogen, 11752050), 1 μg of total RNA was used to synthesize first-strand cDNA using Superscript III reverse transcriptase. The porimin, caspase-3, caspase-1, Cyt-C, Bax, and Bcl-2 mRNA levels were normalized to GAPDH. The PCR reactions were amplified and analyzed by using the Real-Time PCR System (Applied biosystems, StepOne Software, USA). Primers sequences were showed in online supplemental Table 2.

## 2.12 Annexin V-FITC/PI and porimin-FITC/PI flow cytometry

Quantification of oncosis cells and apoptotic cells were performed using the porimin-FITC/PI (Bioss) and the Annexin V-FITC/PI detection kits (Keygen Biotech) respectively. All procedures were performed according to the manufacturer's instructions. Detection of apoptotic cells (AnnexinV-FITC-positive, PI-positive), and oncosis cells (porimin-FITC positive, PI-positive) was performed using a FACSCalibur (FXP018, T etrazhengbai Biological, China).

## 2.13 Statistical analysis

Experimental data are presented as means ± SD and analyzed with unpaired two-tailed t-tests for pairwise comparisons. For non-normal distributions, medians (IQR) are reported and

**Table 2. Primers sequences of porimin, caspase-3, caspase-1, Cyt-C, Bax, and Bcl-2.**

| Target Name | Primer | |
|---|---|---|
| porimin | F | ATGCTCAGGGCTCTCCAAGTG |
| | R | AAGACGGCTTCACAGTGCTA |
| caspase-3 | F | AGCACCTGGTTACTATTCCTG |
| | R | TAAATTCTAGCTTGTGCGCGTA |
| caspase-1 | F | GTTTTGCCCTTTAGAAATAGCC |
| | R | ACTCTCCGAGAAAGATGTTG |
| Cyt-C | F | AAAAGGAGGCAAGCATAAGACTG |
| | R | ACCTTTGTTCTTGTTGGCAT |
| Bax | F | AGGGTTTCATCCAGGATCGAGCA |
| | R | CAGCTTCTTGGTGGACGCATC |
| Bcl-2 | F | ACTTCTCTCGTCGCTACCGTC |
| | R | CCCCATCCCTGAAGAGTTCCT |

analyzed with the Wilcoxon rank-sum test. Repeated measures are analyzed using repeated measures ANOVA. Survival rates are analyzed using the Kaplan-Meier method and log-rank test. All analyses are performed in GraphPad Prism (version 8.3.0, San Diego, CA, USA) with significance at $P < 0.05$.,

## 3. Results

### 3.1 Exertional heat stroke onset induces rhabdomyolysis development

The EHS group rats were placed in a high-temperature environment simulation chamber with a temperature of $(39.5 \pm 0.5)$°C and humidity of $(55 \pm 5)$% for running exercise. Initially, the rats moved easily, but around 7-10 minutes into the exercise, they began to show signs of fatigue, moving slowly and passively retreating. From 10-12 minutes, some rats started to exhibit instability and would stop in the rest area;they were gently driven to continue exercising. By 15-23 minutes, even with driving, the rats could no longer continue the exercise. After removal, their body temperature was measured at over 43°C, with significant weight loss, lying on the ground unable to right themselves, exhibiting positive righting reflexes, remaining motionless, convulsing, or even falling into a coma. The average heat exposure time T1 for the heatstroke rats was $(19.00 \pm 3.16)$ minutes, with the successful model having a core temperature Tc of $(43.82 \pm 0.57)$°C. Before and after modeling, the rats were weighed;the average pre-modeling weights of the CN group and EHS group were similar, but post-modeling, the EHS group rats showed a noticeable decrease in weight, with the weight loss accounting for approximately $(7.87 \pm 1.54)$% of their total body weight, which is 4.1 times greater than the weight loss of the CN group rats (Table 3).

To elucidate the mechanism underlying the development of RM, we have developed an EHS rat model with a high propensity for RM, which may truly reflect the clinical features. Our observations revealed a substantial elevation in CK and MB levels following the onset of EHS, with peak concentrations observed at 6 hours post-EHS. Consequently, we selected the 6-hour post-EHS time point for further examination and assessed the extent of skeletal muscle injury across three distinct muscle types: transversus abdominis muscle, musculi soleus, and gastrocnemius (Gas). Notably, histological analysis at 6 hours post-EHS demonstrated the most severe alterations in the Gas muscle, characterized by a significant reduction in muscle fiber density, a markedly disordered muscle fiber arrangement, and pronounced swelling degeneration. The ultrastructure of the gastrocnemius muscle of EHS rats showed muscle

**Table 3. Body temperature, body weight, and modeling time of heat stroke rats.**

| ggroup | Weight (g) | Tc0 (°C) | TcEHS (°C) | Percent Weight change (%) | Duration of EHS protocol (min) |
|---|---|---|---|---|---|
| CN | 306.40 ± 13.14 | 38.45 ± 0.36 | 38.55 ± 0.27 | 1.90 ± 0.70 | 20 |
| EHS | 301.80 ± 15.44 | 38.55 ± 0.31 | 43.82 ± 0.57 | 7.87 ± 1.54 | 19.00 ± 3.16 |
| P | 0.733 | 0.468 | <0.001 | <0.001 | 0.457 |

fiber swelling, disordered structure, loss of Z-line, destruction of sarcomere, vacuolization of cytoplasm, obvious swelling and degranulation of mitochondria, disappearance of mitochondrial cristae, and nuclear degeneration. These findings suggest that the Gas muscle sustains the most severe damage following EHS.

In the present study, we reproduced the EHS rat model, which showed exhaustion, absent righting reflex, lying still, listlessness, convulsions and other mental symptoms, and Tc was greater than or equal to 43°C, and weight loss was about 7.87 ± 1.54% (Table 3). we observed that the CK (from 323.7 ± 14.29 U/L to 2352 ± 149.2 U/L) and MB (from 12.13 ± 0.74U/L to27.99 ± 0.35 U/L)levels were significantly increased, and Gas muscle injury at 6h after EHS. The 6-hour survival rate of EHS rats was significantly lower than that of controls (Fig 1).

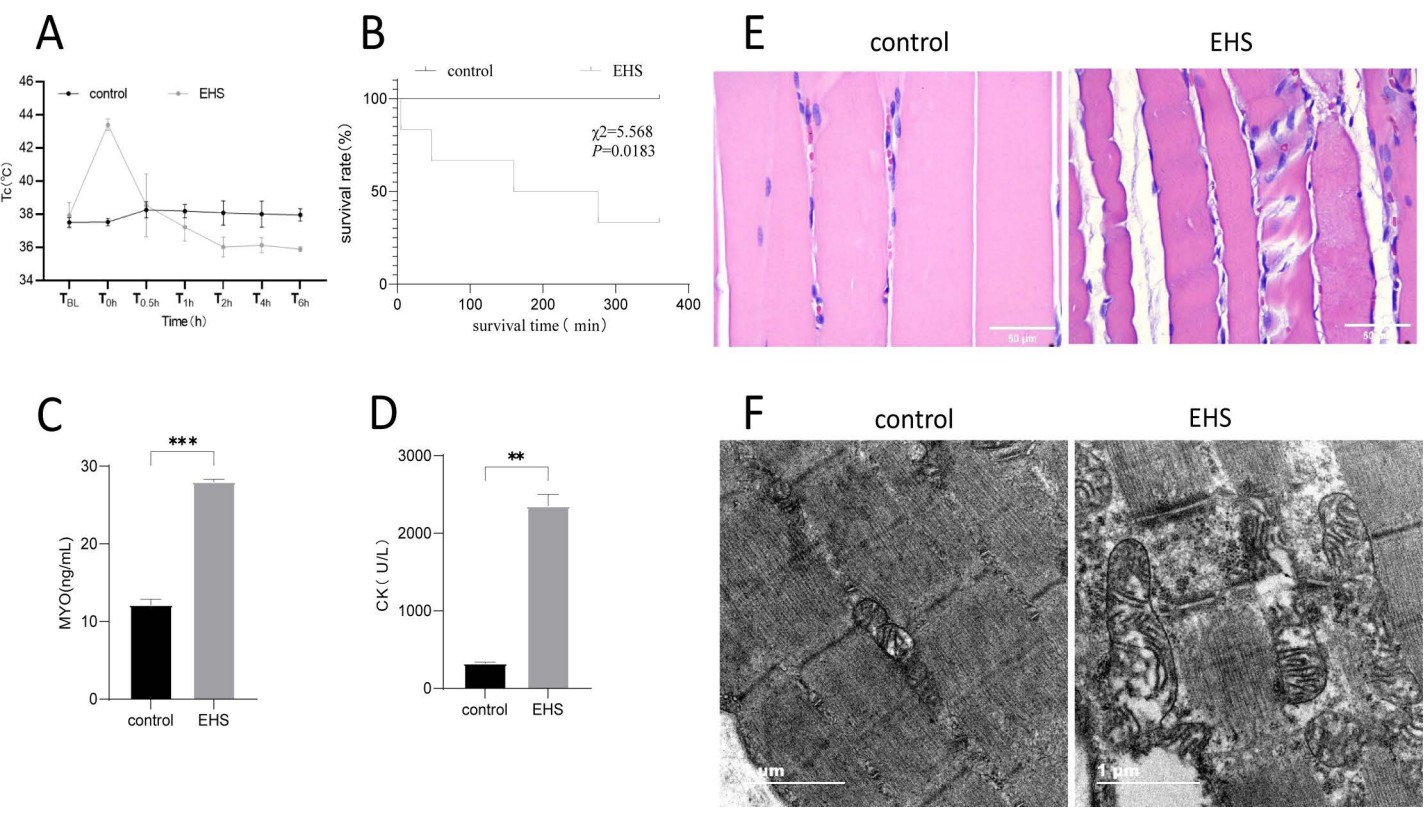

**Fig 1. Exertional heat stroke onset induces rhabdomyolysis development.** (A) Trends in body temperature in rats with exertional heat stroke ($n$ = 6). (B) Kaplan-Meier survival curves for EHS rats ($n$ = 6). (C-D) Serum creatine kinase (CK) levels and MB levels were measured at 6 hours after the onset of EHS ($n$ = 6). (E) Representative gastrocnemius muscle H&E staining (CN and EHS, ($n$ = 6), scale bar in H&E = 50μm). (F) Ultrastructural changes of gastrocnemius muscle in exertional heat stroke rats under electron microscope (CN and EHS, ($n$ = 6), scale bar = 1μm).**, $P$ < 0.01;***, $P$ < 0.001.

## 3.2 The pathological mechanism of rhabdomyolysis in exertional heat stroke

In order to investigate the death mode of skeletal muscle cells in RM following EHS, we examined and analyzed from multiple perspectives such as inflammation, stress, apoptosis, and oncosis.

We first focused on the contribution of oncosis and apoptosis to rhabdomyolysis development following EHS.The ultrastructure of the gastrocnemius muscle of EHS rats showed muscle fiber swelling, larger volume, disordered structure, loss of Z-line, destruction of sarcomere, vacuolization of cytoplasm, obvious swelling and degranulation of mitochondria, disappearance of double cristas, and nuclear degeneration (Fig 1). Porimin (pro-oncosis receptor inducing membrane injury), Pro-oncosis receptor inducing membrane injury, had been shown that porimin is specifically expressed on the surface of cells that are about to oncosis [9]. In this study, porimin membrane protein, consistently confirmed by WB, immunohistochemical staining and qPCR, was highly expressed in the gastrocnemius muscle of EHS rats (Fig 2). However, there was no significant change in caspase-3 in rats of the EHS group compared with the control group (Fig 2). These results suggest that oncosis may be one of the important mechanisms of RM. In addition, WB and qPCR results showed that the expression of pro-apoptotic protein Bax was increased and the expression of anti-apoptotic protein Bcl-2 was decreased in the gastrocnemius muscle of rats in the EHS group, suggesting that apoptosis was also involved in the occurrence of rhabdomyolysis related to heat stroke.

In addition, we also detected NLRP3 and caspase-1 by WB and immunohistochemistry (Fig 2), and reactive oxygen species (ROS) by fluorescent probe DCFH-DA. The results showed that the expression of NLRP3 and caspase-1 in the gastrocnemius muscle of rats in the EHS group increased, and ROS production increased. These results suggest that oxidative stress and inflammation are also involved in rhabdomyolysis.

## 3.3 Oncosis, rather than apoptosis, is the main form of skeletal muscle cell death

To determine the main mechanism of rhabdomyolysis, HSKMC cells were exposed to 43°C water heat stress for 4 hours to establish a heat stroke rhabdomyolysis cell model. The results of light microscopy and CCK8 showed that the damage of HSKMC cells gradually deepened despite the recovery of normal temperature. At 24 hours after resuscitation, the cells showed patchy necrosis, and at 48 hours, the cells showed large necrosis, and the cell viability was less than 50%. Therefore, we selected 24 hours of resuscitation as the observation time point (Fig 3A-3B).

Our results (Fig 3C) showed that after 24 hours of resuscitation, HSKMC cells showed oncosis-like day shift morphology, such as cell swelling, larger volume, cytoplasmic vacuolization, obvious swelling and degranulation of mitochondria, disappearance of double cristae, swelling of nucleus, and interruption of partial envelope continuity. In the later stage (oncosis-like necrosis), the nucleolus was dissolved and disappeared, the nucleus was fragmented, the chromatin was gathered, and the cytoplasmic structure was disintegrated and granular. Flow cytometry (Fig 3H-3K)showed that the number of double positive cells (porimin+/PI+) in HS group increased significantly ( $(21.36 \pm 1.16)$% vs $(2.61 \pm 0.29)$%, $P < 0.001$). However, the number of Annexin V+/PI+ double positive cells was not significantly increased ($P \geq 0.05$). In addition, WB results (Fig 3D-3G)showed that the expression of porimin protein in HSKMC cells of HS group was significantly increased, while the expression of caspase-3 protein was not significantly changed. These results strongly suggested that oncoptosis rather than apoptosis was the main form of death of HSKMC cells after heat stress.

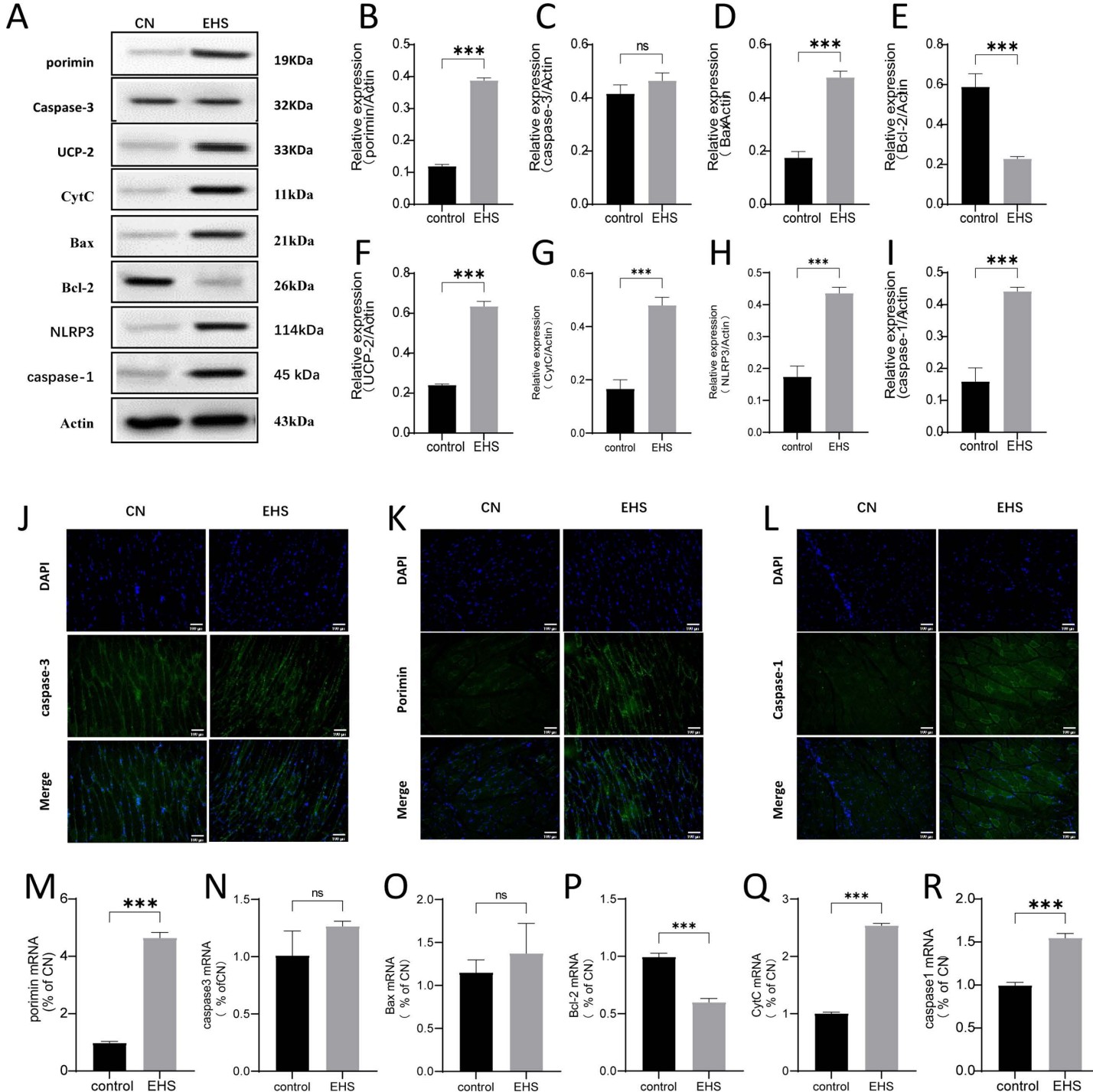

**Fig 2. Apoptosis, oxidative stress, oncosis and other mechanisms are involved in the pathological mechanism of rhabdomyolysis in exertional heat stroke.** (A) Western blot was used to detect the expression levels of porimin, caspase-3, UCP-2, CytC, Bax, Bcl-2, NLRP3, and caspase-1 in EHS rats ($n = 6$). (B-I) Quantitative analysis of porimin, caspase-3, UCP-2, CytC, Bax, Bcl-2, NLRP3, and caspase-1 protein expression levels in gastrocnemius muscle of CN group and EHS group, respectively, ($n = 6$). (J-L) Representative images of the immunofluorescence detection of caspase-3, porimin and caspase-1 respectively in the gastrocnemium of rats in each group (CN and EHS, $n = 6$, scale bar = 100μm). (M-R) The relative expression levels of porimin, caspase-3, UCP-2, CytC, Bax, Bcl-2, NLRP3, and caspase-1 mRNA in the gastrocnemius muscle of the EHS group were quantitatively analyzed compared to those of the CN group, ($n = 6$). ns, $P \geq 0.05$; *, $P < 0.01$; **, $P < 0.01$; ***, $P < 0.001$.

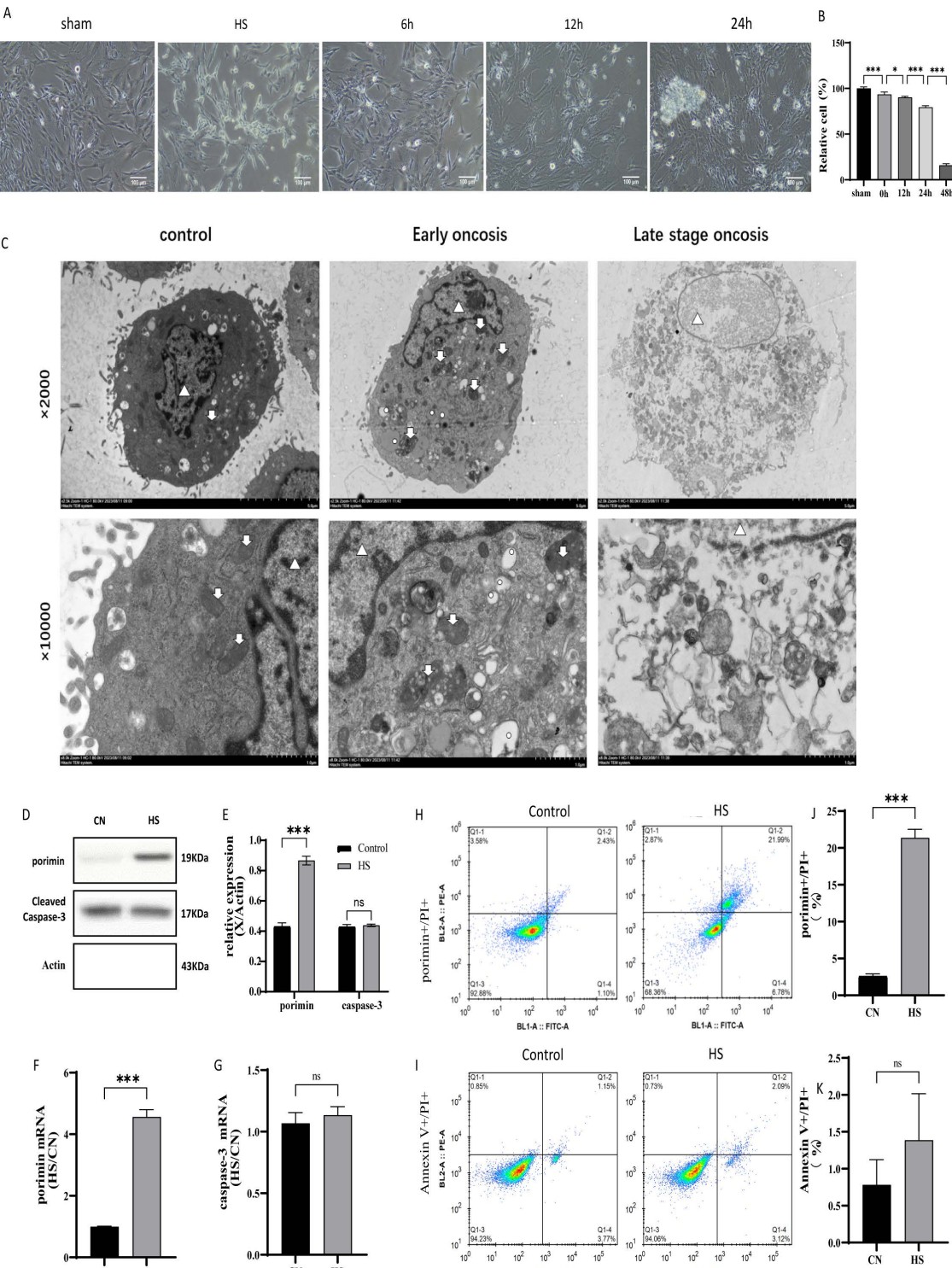

**Fig 3. Oncosis, rather than apoptosis, is the main form of skeletal muscle cell death.** (A) Morphological changes in the recovery phase of HSKMC cells after 4 hours of heat stress in a 43 ° C water bath. (CN and HS, scale bar = 100μm) (B) Changes in cell viability of HSKMC during the recovery phase after heat stress. (C) Effect of heat stress on ultrastructural changes in HSKMC cells. (D) WB representative images of porimin and cleaved caspase-3 proteins in HSKMC cells exposed to 4-h heat stress in 43 ° C water and then revived for 24h; (E) Quantitative analysis of relative porimin and cleaved caspase-3 protein expression levels. (F-G) Quantitative analysis of relative expression levels of porimin mRNA and caspase-3 mRNA in HS group compared to those of the CN group. (H-I) Flow cytometry was used to detect the rate of porimin + /PI + double positive cells and Annexin V + /PI + double positive cells.

(J-K) Quantitative analysis of (porimin+/PI+) double positive cell rate and Annexin V+/PI+ double positive cell rate in HSKMC cells of HS group and CN group. ns, $P \geq 0.05$; *, $P < 0.01$; **, $P < 0.01$;***, $P < 0.001$.

### 3.4 Mitochondrial damage is a key event in the mechanism of oncosis in skeletal muscle cells

Given that an energy deficiency is the primary cause for oncosis [12]. In our in vivo and in vitro experiments, we have thoroughly evaluated the changes in mitochondrial structure and function. Transmission electron microscopy analysis reveals that the mitochondria in the EHS mouse gastrocnemius muscle are significantly swollen, with increased volume, blurred internal structures, and incomplete structures in some mitochondria (Fig 1). JC-1 staining for mitochondrial membrane potential shows reduced red fluorescence and increased green fluorescence, indicating a decrease in mitochondrial membrane potential (Fig 4A-4B). Additionally, ATP assay kit detection shows a significant decrease in ATP levels in the EHS group's gastrocnemius (Fig 4C). Western blot analysis reveals an increased CytC content in the gastrocnemius muscle of EHS group (Fig 3).

In vitro experiments also confirm these findings, as HS group HSKMC cells show significant mitochondrial swelling, loss of cristae, and decreased ATP and ROS levels after heat shock for 4 hours and recovery for 24 hours (Fig 4D-4F). These results suggest that the mitochondrial structure and function in EHS mouse colon and heat-shocked HSKMC cells are damaged, indicating that mitochondrial damage is a major mechanism of cell death.

### 3.5 ROS regulated porimin-dependent oncosis

Reactive oxygen species (ROS) have a close relationship with mitochondrial damage, and they play a key role in oncosis [10,16].We conducted combined in vivo and in vitro experiments to detect ROS levels in rat gastrocnemius muscle and HSKMC cells. Using the DCFH-DA fluorescent probe, we observed significantly enhanced green fluorescence and increased ROS content in both the EHS (exercise-heat stress) rat model gastrocnemius muscle and the HS (heat stress) group HSKMC cells compared to their respective controls (Fig 5).

These findings suggest that ROS levels are significantly increased in both in vivo and in vitro models of oncosis, indicating a close relationship between ROS and mitochondrial damage. The increase in ROS levels may play a critical role in the development of oncosis. Further studies are needed to investigate the underlying mechanisms and potential therapeutic targets.

## 4. Discussion

In our study, we have identified that mitochondrial damage is a critical event in the oncosis mechanism of muscle cell death in skeletal muscle cells. Oncosis, also known as cell swelling, is a form of regulated cell death characterized by a rapid loss of plasma membrane integrity, cell swelling, and eventual rupture.

Current research on oncosis is still insufficient both domestically and internationally, with the detection of oncosis being a particularly challenging issue. Porimin is a specific marker for oncosis, and cell and organelle swelling is the primary morphological change in oncotic cells. The currently recognized methods for detecting oncosis include electron microscope ultrastructure and Porimin protein detection [10,15,17–19]. Flow cytometry has been used in many studies on oncosis, although it is mostly used with Annexin V/PI double staining and opinions are divided. Some researchers consider oncotic cells to be Annexin V+/PI+ double-positive cells [20,21], while others argue that oncotic cells are Annexin V+/PI- or Annexin V-/PI+ [22]. Some researchers consider both Annexin V+/PI+ and Annexin V-/PI+ cells to be oncotic cells [14,23]. Liu et al. found that the number of membrane-associated

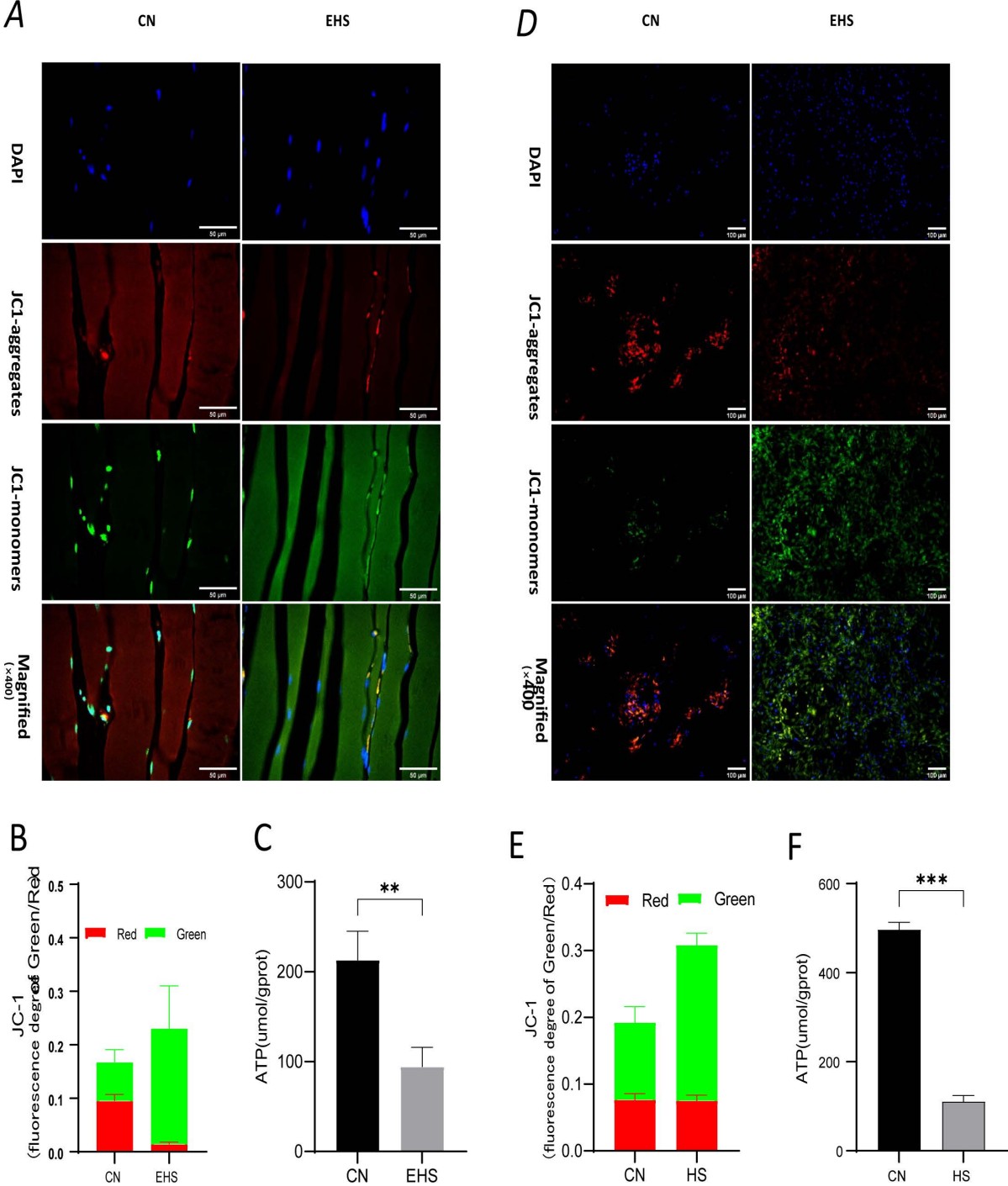

**Fig 4. Mitochondrial damage is a key event in the mechanism of oncosis in skeletal muscle cells.** (A-B) Effect of EHS on the membrane potential of gastrocnemius muscle in rats (CN and EHS, n = 6, scale bar = 50μm). Fresh frozen sections were taken 6 hours after the onset of EHS to detect mitochondrial membrane potential in muscle tissue by JC-I staining. DAPI was used for nuclear staining and is visible in blue, J-monomer lines represent impaired mitochondrial membrane potential (visible in green) and J-aggregates represent normal mitochondrial membrane potential (visible in red/orange). (C) Effect of EHS on ATP in rat gastrocnemius muscle (n = 6). (D-E) Effects of heat stress on mitochondrial function and membrane potential in HUKMC cells (CN and HS, scale bar = 100μm). (F) Effect of heat stress on intracellular ATP levels in HSKMC. **, $P < 0.01$; ***, $P < 0.001$.

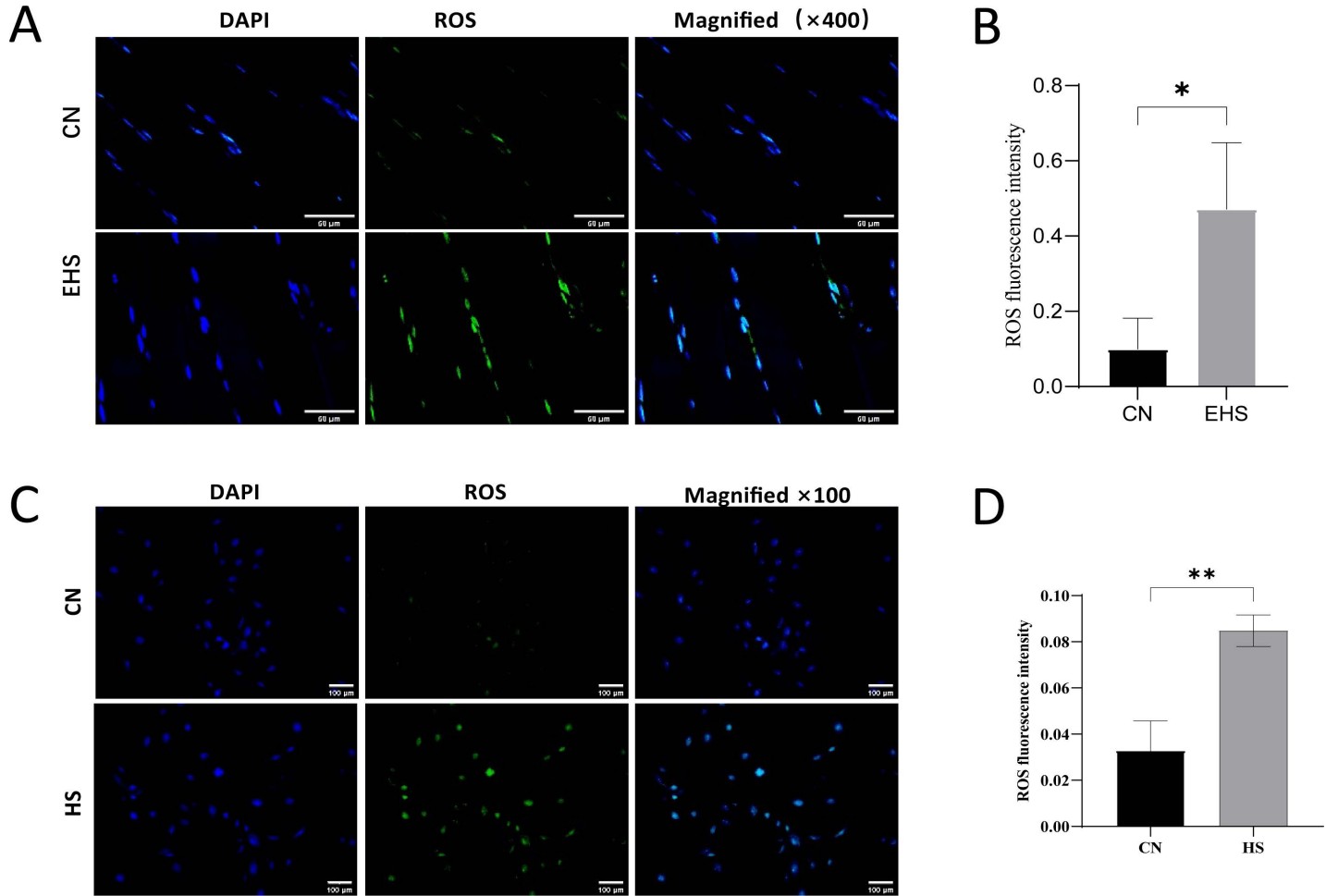

**Fig 5. ROS regulated porimin-dependent oncosis.** (A-B) Effects of EHS on ROS levels in the gastrocnemius of rats (CN and EHS,n = 6, scale bar = 50μm). (C-D) Effect of heat stress on intracellular ROS levels in HSKMC (CN and HS, scale bar = 100μm).

protein V-positive cells (less than 5% of MDA-MB-231 shATG12 cells, less than 12% of H1299 shATG12 cells) cannot reflect a large amount of cell death, and therefore consider these cells to be oncotic cells [17]. Enrique et al. [23] used flow cytometry to quantify the relative cell size and complexity, and analyzed the form of cell death through forward (cell size) and side scatter (cell complexity) plots (FSC vs SSC), with an increase in FSC/SSC representing a population of swollen cells, and a low FSC/SSC ratio indicating a population of small, granular apoptotic and dead cells. JC-1 detection of mitochondrial membrane potential reduction and ATP kit detection of ATP content reduction are common methods for detecting oncosis, but they lack specificity and specific values, and are therefore mostly used as auxiliary detection methods [17,24–28]. Additionally, there are studies that detect intracellular $Ca^{2+}$ levels to detect oncotic cells [17,21].

It is well known that Annexin V-FITC/PI double staining flow cytometry is a classic method for detecting apoptosis, with Annexin V + /PI- cells considered early apoptotic cells, Annexin V + /PI + cells considered late apoptotic cells, and Annexin V-/PI + cells considered necrotic cells. However, due to differing opinions on the use of Annexin V-FITC/PI double staining flow cytometry to identify oncotic cells, and given that Porimin protein is a recognized marker for oncotic cell membranes, we believe that (Porimin + /PI+) double-positive

cells are oncotic cells. By using flow cytometry to detect the rate of (Porimin + /PI+) double-positive cells and comparing it with the rate of (Annexin V + /PI+) double-positive cells, we can compare the roles of oncosis and apoptosis in the muscle fiber degeneration and necrosis of heat shock injury, and our results show that oncosis, rather than apoptosis, is the main form of muscle fiber degeneration and necrosis in heat shock injury.

Mitochondrial damage can occur due to various factors, including calcium overload, reactive oxygen species (ROS) production, and mitochondrial permeability transition (MPT). Calcium overload can result from the failure of ATP-dependent ion pumps, such as $Na^+/K^+$-ATPase and $Ca^{2+}$-ATPase, leading to an accumulation of $Ca^{2+}$ in the mitochondria. This $Ca^{2+}$ overload can activate MPT, which is characterized by a sudden increase in the permeability of the mitochondrial membrane. MPT can result in the depolarization of the mitochondrial membrane potential, release of cytochrome C, and activation of caspase, a key protein involved in cell death.

On the other hand, mitochondria are the primary sources of ATP and ROS, and their damage can lead to ATP depletion and ROS production. ROS production can also cause mitochondrial damage by oxidizing mitochondrial components, such as lipids, proteins, and DNA. Mitochondrial damage can further exacerbate ATP depletion and ROS production, creating a vicious cycle. In addition, ROS can also activate UCP2. Recent studies have shown that Increased ROS levels decrease mitochondrial membrane potential and ATP levels, and that dihydrotanshinone triggers porimin-dependent oncosis by ros-mediated mitochondrial dysfunction in non-small-cell lung cancer [10].

Uncoupling protein 2 (UCP-2) is a mitochondrial anion carrier protein that regulates ATP and ROS production. Its main function is to regulate the proton leakage on the mitochondrial inner membrane, thereby reducing the mitochondrial membrane potential ΨM. The functions of UCP-2 include reducing the release of superoxide anions, negatively regulating the intracellular ROS level, and regulating the intracellular ATP level.

Appropriate increase in UCP-2 expression will lead to a rapid decrease in mitochondrial membrane potential, which will reduce the generation of NADH in mitochondria, thereby reducing the intracellular ATP level [29]. The up-regulation of UCP-2 will also lead to the loss of mitochondrial membrane potential, which is related to the process of oncosis.

Our experiments have revealed significant mitochondrial damage, including mitochondrial swelling and loss of cristae, as observed under the electron microscope. JC-1 staining has also indicated mitochondrial membrane potential damage, suggesting mitochondrial dysfunction. Additionally, we have observed a significant decrease in ATP levels in the mitochondria, expression levels of elevated UCP2 levels, and a marked increase in ROS production in the muscle cells.

Therefore, we propose that the mechanism of muscle cell death in skeletal muscle cells undergoing oncosis is primarily due to ATP depletion and mitochondrial damage. Furthermore, ROS production is both a cause and consequence of mitochondrial damage, further exacerbating muscle cell death. In conclusion, our study highlights the critical role of mitochondria in the regulation of oncosis and provides a potential therapeutic target for preventing muscle cell death in skeletal muscle disorders.

The mechanisms underlying muscle cell death during oncosis have been a subject of considerable interest. Our study concludes that the primary cause of muscle cell death in skeletal muscle cells undergoing oncosis is attributed to ATP depletion and mitochondrial damage. This conclusion is supported by our findings and aligns with existing literature highlighting the critical role of ATP in maintaining cellular homeostasis.

One of the intriguing questions raised by this finding is why these alterations lead to oncosis rather than apoptosis. To address this, we propose several potential mechanisms: (1) ATP Depletion and Plasma Membrane Dysfunction.ATP depletion is a primary factor initiating oncosis. Rapid ATP depletion leads to the inactivation of the $Na^+/K^+$-ATPase pump on the

plasma membrane. This inactivation results in increased intracellular sodium ( [Na$^+$]i) and chloride ( [Cl$^-$]i) concentrations. Consequently, water influx occurs, leading to cellular swelling and an increase in intracellular calcium ( [Ca2+]i) [30]. These changes collectively contribute to the characteristic features of oncosis, including membrane rupture and organelle dysfunction [30]. (2) Energy Crisis and Apoptosis Inhibition.Apoptosis requires ATP and the caspase system. Caspases, such as caspase-3, play a crucial role in apoptosis by cleaving poly (ADP-ribose) polymerase (PARP) [31], a key enzyme involved in DNA repair and maintenance of genomic integrity. When DNA damage is severe, PARP activation facilitates apoptosis [32]. However, when ATP levels are critically low, the regeneration of ADP-ribose polymers required for PARP function becomes compromised. Furthermore, excessive activation of PARP can lead to the depletion of NAD + and ATP, which are essential for the completion of apoptosis. This energy crisis favors the progression towards oncosis instead of apoptosis [33].

### Innovations

Improved Cell Modeling Conditions: We refined the heat shock conditions to a 43°C water bath for 4 hours, effectively creating a model of heat stroke-induced rhabdomyolysis in muscle cells. This approach is more efficient and stable compared to the previous method of a 43°C incubator for 2 hours.

**In Vivo validation.**   Currently, there is limited research on tumor dissolution in heat stroke-related rhabdomyolysis both domestically and internationally. Our in vivo experiments validate the role of tumor dissolution, offering new directions for research and treatment of rhabdomyolysis.

**Unified detection methods**:  There are varying methods to verify tumor dissolution in cells, lacking a standardized approach. Based on Porimin protein being a recognized marker for tumor dissolution, we propose that Porimin + /PI + double-positive cells indicate tumor dissolution. By using flow cytometry to measure the proportion of Porimin + /PI + double-positive cells and comparing it with Annexin V + /PI + double-positive cells, we found that tumor dissolution, rather than apoptosis, is the primary cause of muscle fiber degeneration and necrosis in heat shock injury.

## 5. Summary

This study constructed rat models of exertional heatstroke with rhabdomyolysis both in vivo and in vitro to investigate cell death mechanisms. In vivo, oncosis was identified as a key factor, linked to mitochondrial dysfunction and ATP depletion due to ROS damage and UCP-2 activity. In vitro, oncosis, not apoptosis, was confirmed as the primary cell death mechanism, driven by increased ROS, mitochondrial damage, and ATP depletion.

This study shows that ATP depletion and mitochondrial dysfunction are critical in muscle cell death during rhabdomyolysis, with oncosis being the main form of cell death. Future research should focus on the signaling pathways causing ATP depletion and explore therapies to enhance ATP levels or protect mitochondria. These findings may lead to new treatments for rhabdomyolysis and related conditions.

### Acknowledgements

The authors would like to thank Dr. Shan Yi (Department of Emergency Medicine, PLA Sixth Medical Center) and Dr. Liu Shuyuan (Department of Emergency Medicine, PLA Sixth Medical Center) for advice on experimental design and language polishing. We also thank Dr Mao Handing (Department of Emergency Medicine, PLA Sixth Medical Center) and Dr Yang Wenjun (Department of Emergency Medicine, PLA Sixth Medical Center) for his help in evaluating the pathological images.

## Author contributions

**Conceptualization:** Chengcheng Li, Wenjun Yang, Handing Mao, Yi Shan.

**Data curation:** Chengcheng Li, Wenjun Yang.

**Formal analysis:** Chengcheng Li, Wenjun Yang, Handing Mao.

**Funding acquisition:** Shuyuan Liu, Yi Shan.

**Investigation:** Yang Liu.

**Methodology:** Chengcheng Li, Yang Liu, Handing Mao.

**Project administration:** Chengcheng Li, Shuyuan Liu, Yi Shan.

**Resources:** Shuyuan Liu, Yi Shan.

**Software:** Chengcheng Li, Wenjun Yang.

**Supervision:** Wenjun Yang, Shuyuan Liu, Yi Shan.

**Validation:** Handing Mao, Yi Shan.

**Visualization:** Chengcheng Li, Wenjun Yang.

**Writing – original draft:** Chengcheng Li, Yang Liu.

**Writing – review & editing:** Chengcheng Li, Yang Liu.

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
