## [Decision Letter · Decision Letter 0]

4 Oct 2024

PONE-D-24-30189Oncosis is the Predominant type of cell death in rhabdomyolysis following Exertional heat strokePLOS ONE

Dear Dr. Shan,

Thank you for submitting your manuscript to PLOS ONE. After careful consideration, we feel that it has merit but does not fully meet PLOS ONE’s publication criteria as it currently stands. Therefore, we invite you to submit a revised version of the manuscript that addresses the points raised during the review process.

We look forward to receiving your revised manuscript.

Kind regards,

Gokhan Burcin Kubat

Academic Editor

PLOS ONE

Journal Requirements:

2. To comply with PLOS ONE submissions requirements, in your Methods section, please provide additional information regarding the experiments involving animals and ensure you have included details on  methods of sacrifice, and  efforts to alleviate suffering.

“The funding support from Military medical Innovation Project (18CXZ019) for this work is gratefully acknowledged (CW).”

6. Please include a separate caption for each figure in your manuscript.

7. PLOS ONE now requires that authors provide the original uncropped and unadjusted images underlying all blot or gel results reported in a submission’s figures or Supporting Information files. This policy and the journal’s other requirements for blot/gel reporting and figure preparation are described in detail at https://journals.plos.org/plosone/s/figures#loc-blot-and-gel-reporting-requirements and https://journals.plos.org/plosone/s/figures#loc-preparing-figures-from-image-files. When you submit your revised manuscript, please ensure that your figures adhere fully to these guidelines and provide the original underlying images for all blot or gel data reported in your submission. See the following link for instructions on providing the original image data: https://journals.plos.org/plosone/s/figures#loc-original-images-for-blots-and-gels.  

8. Please include your tables as part of your main manuscript and remove the individual files. Please note that supplementary tables (should remain/ be uploaded) as separate 'supporting information' files

Additional Editor Comments:

As academic editor, I evaluated the reviewers' comments and decided that the manuscript needed major revisions. You can find the reviewers' comments below.

Reviewers' comments:

Reviewer's Responses to Questions

**Comments to the Author**

1. Is the manuscript technically sound, and do the data support the conclusions?

Reviewer #1: No

Reviewer #2: Yes

Reviewer #3: Partly

Reviewer #4: Yes

2. Has the statistical analysis been performed appropriately and rigorously? 

Reviewer #1: No

Reviewer #2: Yes

Reviewer #3: Yes

Reviewer #4: Yes

3. Have the authors made all data underlying the findings in their manuscript fully available?

Reviewer #1: No

Reviewer #2: Yes

Reviewer #3: Yes

Reviewer #4: Yes

4. Is the manuscript presented in an intelligible fashion and written in standard English?

Reviewer #1: Yes

Reviewer #2: Yes

Reviewer #3: Yes

Reviewer #4: Yes

5. Review Comments to the Author

Reviewer #1: The authors find that EHS rhabdomyolysis is associated with oncosis. However, there are major concerns that reduce the reliability of the study results and enthusiasm for the work. First, the novelty of this work is unclear. In the introduction, the authors suggested that evidence already exists "the mechanism and morphological changes of rhabdo exhibit striking similarities to those observed in oncosis". Second, is this a study of EHS or rhabdo? It is known that only a small portion of EHS patients are associated with rhabdo. It seemed that all the RM rats suffered a heat stroke. Finally, my biggest concern is an insufficient sample size that supports the conclusions. As described, this study included 12 rats, and they were divided into CN and EHS groups (6 per group). Fig 1B indicates that 4 of the 6 rats were lost before 6 hr post-EHS. The animal samples were obtained 6 hrs after EHS. So only 2 rats were included in the EHS group for comparison.

Abstract

ehsRM was defined twice. The conclusions are confusing.

Introduction

Some statements are misleading. For example, there is no evidence for "Oncosis, a form of porimin-dependent non-programmed cell death, is caspase independent..". 

Results

In addition to mean/SD, individual values should be displayed in all the bar graphs

3.1: How do you know "onset"? There are no additional data other than that at  6 hr post-EHS. 

Should this be Table 3? why did CN rats lose weight as well?

3.2: oxidative stress. There are no OS makers in Fig 2

The authors should consider having this manuscript professionally edited.

Reviewer #2: This manuscript investigates the role of oncosis as the primary mechanism of cell death in exertional heat stroke-associated rhabdomyolysis (ehsRM). While the study presents a compelling argument supported by in vitro and in vivo experiments, there are several areas that could be improved or further elaborated. Below are specific comments

1.Mechanistic Link Between Mitochondrial Dysfunction and Oncosis: While the study documents mitochondrial damage and links it to oncosis, the exact molecular mechanisms connecting mitochondrial dysfunction and oncosis could be elaborated further. How mitochondrial swelling directly contributes to oncosis versus other forms of necrosis could be clarified, especially in relation to the release of DAMPs (damage-associated molecular patterns) or mitochondrial permeability transition (MPT).

2.Involvement of Mitochondrial Dynamics: Mitochondrial dynamics (fusion, fission, mitophagy) are critical for maintaining mitochondrial integrity, especially under stress conditions. The manuscript does not address whether there are any disruptions in mitochondrial dynamics that could contribute to the observed mitochondrial swelling and dysfunction. Including markers of fusion (e.g., MFN2, OPA1) and fission (e.g., DRP1) could provide a more comprehensive understanding of how mitochondria fail during ehsRM.

3.Incomplete Exploration of ROS-Mediated Mitochondrial Damage: Although the study demonstrates increased ROS levels and mitochondrial dysfunction, it does not fully explore the interplay between ROS production and mitochondrial damage. Mitochondria are both a source and target of ROS. An investigation into whether mitochondrial ROS production exacerbates the observed dysfunction would be valuable, possibly by using mitochondrial-targeted antioxidants in additional experiments.

4.Mitochondrial Bioenergetics: The study primarily measures ATP levels, but a more detailed analysis of mitochondrial bioenergetics, including the activity of key respiratory chain complexes (e.g., Complex I-IV), would strengthen the conclusions regarding mitochondrial dysfunction.

Recommended Additional Experiments

1.Mitochondrial Dynamics: Assess the expression of proteins involved in mitochondrial fusion (MFN1/2, OPA1) and fission (DRP1, FIS1). This could help elucidate whether altered mitochondrial dynamics contribute to the observed mitochondrial swelling and dysfunction in ehsRM.

2.Mitochondrial Respiration: Perform mitochondrial respirometry to quantitatively assess mitochondrial respiratory function, including oxygen consumption rate (OCR), basal respiration, ATP production, and maximal respiratory capacity in the EHS model.

3.Mitochondrial ROS Scavengers: Introduce mitochondrial-targeted antioxidants to determine whether reducing mitochondrial ROS levels can alleviate mitochondrial damage and oncosis. This could provide a therapeutic angle to prevent or mitigate cell death in ehsRM.

4. Ambiguity in Statistical Presentation: The manuscript does not consistently report statistical significance (e.g., p-values) across experiments. Additionally, error bars on graphs are not always explained (e.g., standard deviation vs. standard error). Ensure that all statistical analyses are clearly reported, with appropriate p-values and error bars labeled. Use more rigorous statistical tests where appropriate.

5. Lack of Functional Outcomes or Behavioral Correlates in Vivo: The manuscript focuses heavily on molecular and structural analysis but lacks a functional or clinical correlate for rhabdomyolysis and EHS severity in the rat model. Behavioral or functional assessments of muscle function (e.g., grip strength, endurance) post-EHS would help translate these findings into clinically relevant outcomes. Include behavioral or physiological assessments in the rat model to assess muscle function after EHS and correlate them with biochemical and histopathological findings.

6. Limited Validation of Oncosis Markers: While the study highlights porimin as a marker for oncosis, the absence of multiple independent markers limits the specificity of the conclusions. Porimin alone may not provide conclusive evidence of oncosis as the dominant mechanism, especially given the presence of apoptotic markers (Bax/Bcl-2). Incorporate additional markers (e.g., HMGB1, or others associated with oncosis) and perform colocalization experiments with mitochondrial damage markers to reinforce the claim of oncosis.

7.Unclear Differentiation between Oncosis and Apoptosis: The manuscript claims that oncosis is the predominant form of cell death, but the presence of apoptotic indicators such as Bax and Bcl-2 raises questions. Although caspase-3 levels did not change, the manuscript does not adequately explain why apoptosis, in addition to oncosis, is not more prominent.Conduct time-course experiments to clearly differentiate the dynamics of oncosis and apoptosis. A detailed analysis of caspase-independent apoptotic pathways could be beneficial to clarify this distinction.

Reviewer #3: 1. Introduction and discussion section should be improved.

The difference between apoptosis and oncosis should be described in detail. Authors should emphasize the importance of this study? Is it really valuable to discriminate oncosis and apoptosis?

2. EHS protocol is stopped when the animals become exhausted. Please give the total duration of protocol for each animal.

3. Authors have provided sterile water. What about chow? Is it also sterile?

4. Authors put cell lines into a hot water bath. How did they gassed the cells? Did authors also consider the solubility change between 37 and 43 °C

5. 30 minutes is a relatively long centrifuge time. Why did authors prefer a long centrifuge time?

6. Authors measured CK. Is it total? Were they able to identify subtypes?

7. Could authors cite the previous study? The study in which they first made the EHS model.

8. In Figure 1: Annexin bars seem relatively different on the other hand your standart deviation is very high. That's why you couldn't observe statistical significance. Do you consider to repeat that data?

9. In Figure 4: There is no mitochondrial membrane potential bar chart. There was only two bar charts for ATP level. Please correct.

10. Figure 5: There is no bar graph for ROS.

11. There is also an increase in Bax BCL levels. How can the authors conclude that there is no apoptosis. And please also give Bax/BCL ratio

12. The survival is very bad in EHS group. How many animals were able to finish the study? Please give “n” for each group.

13. Authors concluded “Therefore, we propose that the mechanism of muscle cell death in skeletal muscle cells undergoing oncosis is primarily due to ATP depletion and mitochondrial damage.” Why this alterations caused oncosis rather than apoptosis? Do authors have any suggestions for mechanism?

Minor

All words in latin should be written in italic.

Reviewer #4: the article presents two models, in vivo and in vitro, to further our understanding of the type of cell death, presumably ischemic and associated with primitive mitochondrial damage, oncosis, in rhabdomyolysis associated with exercise heat stroke. The article is well developed and leads in both models, cellular and animal, to identify mitochondrial damage as the primum movens of the mechanism of oncosis.

The primary damage would result in ATP depletion and mitochondrial damage presumably triggered by free radicals (ROS) acting as the “cause and consequence of mitochondrial damage.”

Mitochondrial damage is definitely related to ROS accumulation, which in turn is triggered by inflammatory damage especially of a chronic nature.

Markers of inflammation were evaluated in the presented experimental mouse model? From the most common, such as C-reactive protein (CRP), erythrocyte sedimentation rate (ESR) and plasma viscosity (PV) in blood, to the more cell-specific, such as pro-inflammatory and anti-inflammatory cytokines?

ROS damage is definitely quantitative; there is a correlation between extent of damage and amount of ROS. Have ROS levels been quantified? Is it possible to show an image of the DCFH-DA fluorescent probe?

6. PLOS authors have the option to publish the peer review history of their article (what does this mean? ). If published, this will include your full peer review and any attached files.

**Do you want your identity to be public for this peer review?** For information about this choice, including consent withdrawal, please see our Privacy Policy .

Reviewer #1: No

Reviewer #2: **Yes: ** Meghnad G Joshi

Reviewer #3: No

Reviewer #4: No

---

## [Decision Letter · Decision Letter 1]

18 Dec 2024

PONE-D-24-30189R1Oncosis is the Predominant type of cell death in rhabdomyolysis following Exertional heat strokePLOS ONE

Dear Dr. Shan,

Thank you for submitting your manuscript to PLOS ONE. After careful consideration, we feel that it has merit but does not fully meet PLOS ONE’s publication criteria as it currently stands. Therefore, we invite you to submit a revised version of the manuscript that addresses the points raised during the review process.

We look forward to receiving your revised manuscript.

Kind regards,

Gokhan Burcin Kubat

Academic Editor

PLOS ONE

Journal Requirements:

Additional Editor Comments :

Editor' Comments: There are some minor revisions to the manuscript.

Reviewer's Comments:

Clarity and Readability: The abstract is rich in data but dense, making it difficult for readers to quickly grasp the key findings and implications. Simplify and organize the content for better flow.

Consistency: There are inconsistencies in spacing (e.g., missing spaces after commas) and formatting (e.g., inconsistent use of abbreviations). Ensure uniformity throughout.

Grammar and Syntax: Certain phrases could be reworded for grammatical correctness and conciseness.

Structure: The abstract lacks a clear division between the background, objectives, methods, results, and conclusions. These sections should be clearly identifiable for better readability.

Introduction: The manuscript effectively introduces exertional heat stroke (EHS) and rhabdomyolysis (RM), but the research hypothesis could be stated more clearly.

Materials and Methods: Detailed descriptions of the experimental design, including EHS modeling and assays, are provided. However, statistical methods could be elaborated for transparency.

Results: Results are well-organized and supported by figures and tables. However, the results section could benefit from a more narrative flow linking findings to the study objectives.

Discussion: The discussion highlights the significance of oncosis but should better integrate findings with existing literature to emphasize novelty.

Conclusion: The conclusion is concise but would benefit from a clearer statement of implications and future directions.

Reviewers' comments:

Reviewer's Responses to Questions

**Comments to the Author**

1. If the authors have adequately addressed your comments raised in a previous round of review and you feel that this manuscript is now acceptable for publication, you may indicate that here to bypass the “Comments to the Author” section, enter your conflict of interest statement in the “Confidential to Editor” section, and submit your "Accept" recommendation.

Reviewer #2: All comments have been addressed

Reviewer #3: All comments have been addressed

Reviewer #4: All comments have been addressed

2. Is the manuscript technically sound, and do the data support the conclusions?

Reviewer #2: Partly

Reviewer #3: Yes

Reviewer #4: Yes

3. Has the statistical analysis been performed appropriately and rigorously? 

Reviewer #2: Yes

Reviewer #3: Yes

Reviewer #4: Yes

4. Have the authors made all data underlying the findings in their manuscript fully available?

Reviewer #2: Yes

Reviewer #3: Yes

Reviewer #4: Yes

5. Is the manuscript presented in an intelligible fashion and written in standard English?

Reviewer #2: No

Reviewer #3: Yes

Reviewer #4: Yes

6. Review Comments to the Author

Reviewer #2: General Comments:

Clarity and Readability: The abstract is rich in data but dense, making it difficult for readers to quickly grasp the key findings and implications. Simplify and organize the content for better flow.

Consistency: There are inconsistencies in spacing (e.g., missing spaces after commas) and formatting (e.g., inconsistent use of abbreviations). Ensure uniformity throughout.

Grammar and Syntax: Certain phrases could be reworded for grammatical correctness and conciseness.

Structure: The abstract lacks a clear division between the background, objectives, methods, results, and conclusions. These sections should be clearly identifiable for better readability.

Introduction: The manuscript effectively introduces exertional heat stroke (EHS) and rhabdomyolysis (RM), but the research hypothesis could be stated more clearly.

Materials and Methods: Detailed descriptions of the experimental design, including EHS modeling and assays, are provided. However, statistical methods could be elaborated for transparency.

Results: Results are well-organized and supported by figures and tables. However, the results section could benefit from a more narrative flow linking findings to the study objectives.

Discussion: The discussion highlights the significance of oncosis but should better integrate findings with existing literature to emphasize novelty.

Conclusion: The conclusion is concise but would benefit from a clearer statement of implications and future directions.

Reviewer #3: Thank you for your editing. I think manuscript is ready for publication after this revision. I have no further concerns.

Reviewer #4: (No Response)

7. PLOS authors have the option to publish the peer review history of their article (what does this mean? ). If published, this will include your full peer review and any attached files.

**Do you want your identity to be public for this peer review?** For information about this choice, including consent withdrawal, please see our Privacy Policy .

Reviewer #2: No

Reviewer #3: No

Reviewer #4: No

---

## [Author Response · Author response to Decision Letter 1]

17 Jan 2025

Comment:

1.We note that the grant information you provided in the ‘Funding Information’ and ‘Financial Disclosure’ sections do not match.

When you resubmit, please ensure that you provide the correct grant numbers for the awards you received for your study.

In addition, please state what role the funders took in the study. If the funders had no role, please state: "The funders had no role in study design, data collection and analysis, decision to publish, or preparation of the manuscript."

Kindly include this amended Funding disclosure statement in your cover letter; we will change the online submission form on your behalf.

Response:

Thank you for bringing the funding information discrepancy to our attention. We apologize for any confusion this may have caused.

The funding support for this work was provided by the Military Medical Innovation Project (18CXZ019). Additionally, we confirm that the funders had no role in the study design, data collection and analysis, decision to publish, or preparation of the manuscript.

We have reviewed and corrected the funding information in both the 'Funding Information' and 'Financial Disclosure' sections to ensure consistency. Please find below the amended funding disclosure statement for inclusion in our cover letter:

"The funding support for this work was provided by the Military Medical Innovation Project (18CXZ019). The funders had no role in the study design, data collection and analysis, decision to publish, or preparation of the manuscript."

We kindly request that you update the online submission form with these corrections.

Thank you for your understanding and assistance.

Comment:

2.We note that you have indicated that there are restrictions to data sharing for this study. PLOS only allows data to be available upon request if there are legal or ethical restrictions on sharing data publicly. For more information on unacceptable data access restrictions, please see http://journals.plos.org/plosone/s/data-availability#loc-unacceptable-data-access-restrictions.

b) If there are no restrictions, please upload the minimal anonymized data set necessary to replicate your study findings to a stable, public repository and provide us with the relevant URLs, DOIs, or accession numbers. For a list of recommended repositories, please see https://journals.plos.org/plosone/s/recommended-repositories. You also have the option of uploading the data as Supporting Information files, but we would recommend depositing data directly to a data repository if possible.

Response:

Thank you for your email and the detailed instructions regarding data sharing.

We confirm that all data supporting the results reported in this manuscript are openly available and there are no restrictions on their sharing. As per your recommendation, we have chosen to upload the data as Supporting Information files.

Thank you for your understanding and assistance.

---

## [Decision Letter · Decision Letter 2]

28 Jan 2025

Oncosis is the Predominant type of cell death in rhabdomyolysis following Exertional heat stroke

PONE-D-24-30189R2

Dear Dr. Shan,

We’re pleased to inform you that your manuscript has been judged scientifically suitable for publication and will be formally accepted for publication once it meets all outstanding technical requirements.

Kind regards,

Gokhan Burcin Kubat

Academic Editor

PLOS ONE

Reviewers' comments:

Reviewer's Responses to Questions

**Comments to the Author**

1. If the authors have adequately addressed your comments raised in a previous round of review and you feel that this manuscript is now acceptable for publication, you may indicate that here to bypass the “Comments to the Author” section, enter your conflict of interest statement in the “Confidential to Editor” section, and submit your "Accept" recommendation.

Reviewer #2: All comments have been addressed

2. Is the manuscript technically sound, and do the data support the conclusions?

Reviewer #2: Yes

3. Has the statistical analysis been performed appropriately and rigorously? 

Reviewer #2: Yes

4. Have the authors made all data underlying the findings in their manuscript fully available?

Reviewer #2: Yes

5. Is the manuscript presented in an intelligible fashion and written in standard English?

Reviewer #2: Yes

6. Review Comments to the Author

Reviewer #2: All comments addressed and manuscript has improved in the present form and suitable for publication.

7. PLOS authors have the option to publish the peer review history of their article (what does this mean? ). If published, this will include your full peer review and any attached files.

**Do you want your identity to be public for this peer review?** For information about this choice, including consent withdrawal, please see our Privacy Policy .

Reviewer #2: No

---

## [Editor Report · Acceptance letter]

PONE-D-24-30189R2

PLOS ONE

Dear Dr. Shan,

I'm pleased to inform you that your manuscript has been deemed suitable for publication in PLOS ONE. Congratulations! Your manuscript is now being handed over to our production team.

Kind regards,

on behalf of

Dr. Gokhan Burcin Kubat

Academic Editor

PLOS ONE